# Automated temporalis muscle quantification and growth charts for children through adulthood

Anna Zapaishchykova [1,2,12], Kevin X. Liu[2], Anurag Saraf[1,2], Zezhong Ye[1,2], Paul J. Catalano[3,4], Viviana Benitez[5], Yashwanth Ravipati [1,2], Arnav Jain[1,2], Julia Huang[1,2], Hasaan Hayat[2,6], Jirapat Likitlersuang[1,2], Sridhar Vajapeyam [5,7], Rishi B. Chopra[2], Ariana M. Familiar [8,9], Ali Nabavidazeh[8,9], Raymond H. Mak [1,2], Adam C. Resnick[8,9], Sabine Mueller [10], Tabitha M. Cooney[5], Daphne A. Haas-Kogan [2], Tina Y. Poussaint[5,7], Hugo J.W.L. Aerts [1,2,11] & Benjamin H. Kann [1,2,12] ✉

Lean muscle mass (LMM) is an important aspect of human health. Temporalis muscle thickness is a promising LMM marker but has had limited utility due to its unknown normal growth trajectory and reference ranges and lack of standardized measurement. Here, we develop an automated deep learning pipeline to accurately measure temporalis muscle thickness (iTMT) from routine brain magnetic resonance imaging (MRI). We apply iTMT to 23,876 MRIs of healthy subjects, ages 4 through 35, and generate sex-specific iTMT normal growth charts with percentiles. We find that iTMT was associated with specific physiologic traits, including caloric intake, physical activity, sex hormone levels, and presence of malignancy. We validate iTMT across multiple demographic groups and in children with brain tumors and demonstrate feasibility for individualized longitudinal monitoring. The iTMT pipeline provides unprecedented insights into temporalis muscle growth during human development and enables the use of LMM tracking to inform clinical decision-making.

Lean muscle mass, a primary component of human body composition, is a key indicator of human health and has been linked to numerous outcomes[1,2]. Sarcopenia, characterized by a loss of lean muscle mass, is associated with malnutrition, aging, chronic disease, physiologic frailty, and death in children and adults[2–7]. In children and adolescents, the mechanisms underlying sarcopenia are not fully understood and may involve genetic, epigenetic, and environmental factors such as a sedentary lifestyle and poor nutrition[8]. In addition, sarcopenia in children is more challenging to define and track than in adults due to differences in physiologic development related to age and puberty, which are variable[9]. In children with illnesses like cancer, sarcopenia, and the associated phenomenon, cachexia, are associated with

[1]Artificial Intelligence in Medicine (AIM) Program, Mass General Brigham, Harvard Medical School, Boston, MA, USA. [2]Department of Radiation Oncology, Dana-Farber Cancer Institute, Brigham and Women's Hospital, Boston Children's Hospital, Harvard Medical School, Boston, MA, USA. [3]Department of Data Science, Dana-Farber Cancer Institute, Boston, MA, USA. [4]Department of Biostatistics, Harvard T.H. Chan School of Public Health, Boston, MA, USA. [5]Dana-Farber/Boston Children's Cancer and Blood Disorders Center, Harvard Medical School, Boston, MA, USA. [6]Michigan State University, East Lansing, MI, USA. [7]Department of Radiology, Boston Children's Hospital, Boston, MA, USA. [8]Children's Hospital of Philadelphia, Philadelphia, USA. [9]University of Pennsylvania, Pennsylvania, USA. [10]Department of Neurology, Neurosurgery and Pediatrics, University of California, San Francisco, USA. [11]Department of Radiology and Nuclear Medicine, CARIM & GROW, Maastricht University, Maastricht, the Netherlands. [12]These authors contributed equally: Anna Zapaishchykova, Benjamin H. Kann. ✉e-mail: Benjamin_Kann@dfci.harvard.edu

decreased functional status, quality of life, toxicity from chemotherapy, shorter time of tumor control[10,11] and survival[9,12,13]. Sarcopenia can also alter the metabolism of cytotoxic and therapeutics for pediatric and adult tumors, which may have implications for dosing and scheduling these medications[14].

Currently, routine assessment of sarcopenia relies on monitoring nutritional status, weight, and body mass index (BMI), which are simple to measure but do not directly measure body composition. The limitations of BMI as a measure of body composition have been highlighted by the American Medical Association[15], which recently adopted a policy clarifying that additional metrics outside of BMI should be used in conjunction with other measures of body composition. Furthermore, BMI may underestimate body fat for those who are overweight with decreased muscle mass, and such *sarcopenic obesity* is a common, life-limiting problem in pediatric cancer survivors[16], patients with type 2 diabetes mellitus[17], chronic liver disease[18] and dyslipidemia[19].

Anthropometrics, such as triceps skinfold and mid-upper arm muscle circumference, have been proposed but have not added substantial clinical value due to low precision and reproducibility[20]. Quantitative methods of body composition measurement, such as dual-energy x-ray absorptiometry (DXA), have shown validity at the population level, but validity at the individual level is less certain[11,21,22]. In addition, they require specialized centers and expertise, additional study visits, resulting in increased costs[23,24], as well as radiation exposure. Diagnostic computed tomography (CT) and magnetic resonance imaging (MRI) have emerged as alternative methods to assess sarcopenia using the cross-sectional area of a single vertebral slice[25] or using quantitative whole-body MRI[26–28]. However, radiation exposure (in the case of CT) and excess time (in the case of whole-body MRI) impede their use in children[29]. Furthermore, the clinical utility of muscle mass quantification relies on the development of standardized normal reference ranges, which are established using substantial data. However, obtaining such data for whole-body MRI is unlikely, which limits the development of standardized reference ranges for MRI-based muscle mass quantification. On the other hand, pediatric brain MRI is done routinely[30–32] in conditions such as cancers and neurodevelopmental disease.

The temporalis muscle, situated within the temporal fossa of the skull, is an established indicator of lean muscle mass[33,34] and is universally visible on routine brain MRI, presenting an attractive opportunity to assess lean muscle mass in children at scale. Recently, a semi-automated, deep learning(DL)-based approach for measuring cross-sectional area (CSA) of the temporalis muscle on MRI was shown feasible in a small cohort of adults[35]. However, there have been no large-scale efforts to develop and validate tools to characterize the temporalis muscle and establish normal reference ranges that would be critical to informing health risks and clinical decision-making.

While temporalis muscle thickness (TMT) presents an attractive means to assess body composition, use in young, developing people requires knowledge of normal growth trajectory and distribution, which is currently unknown. To address this problem and enable the utility of TMT in guiding clinical management, we developed *iTMT*, a deep learning-based pipeline for standardized and reproducible temporalis muscle auto-segmentation and TMT calculation. We used iTMT to generate normal, age- and sex-adjusted TMT growth charts, leveraging an aggregated cohort of 23,876 MRI scans from 13 international sources that provide unprecedented insights into muscle growth through young adulthood. Following rigorous validation and acceptability testing, we evaluated use cases of iTMT in predicting nutritional and metabolic deficiencies in healthy children and those with brain tumors. We made our pipeline publicly available for further evaluation and release an online calculator tool for use by the research and clinical communities.

## Results

### Automated temporalis muscle segmentation and thickness calculation (iTMT)

The iTMT pipeline performs three tasks sequentially: 1) axial slice selection at the superior orbital roof, 2) temporalis muscle segmentation, and 3) TMT calculation. The iTMT pipeline was comparable to human experts for the first stage task of superior orbital roof localization compared to ground truth in terms of superior-inferior variability (healthy cohort pipeline median absolute error (MAE): 2.5 mm [IQR: 1–4.75 mm] vs expert MAE: 2.0 mm [IQR: 1.0–2 mm]; cancer cohort pipeline MAE = 3.0 mm [IQR = 2–5 mm]) vs expert MAE: 1.0 [IQR = 0–3 mm] (Fig. 1B). We further find that craniocaudal localization impact on iTMT measurement is negligible within the expected error margin of the pipeline (±4 mm = 2.73%, see Table S3).

iTMT pipeline segmentation performance in the healthy cohort was excellent with median DSC: 0.84, (95%CI: 0.83–0.85) vs median inter-expert DSC 0.81 (95%CI: 0.79–0.83); and in the brain tumor cohort with median iTMT DSC 0.81 (95%CI: 0.73–0.88) vs median inter-expert DSC 0.81 (95%CI: 0.76–0.85). Results suggest negligible performance degradation in the presence of brain tumors and/or surgical manipulation characteristics of the scans in the brain tumor cohort. iTMT measurement accuracy was comparable to an inter-expert agreement in healthy children (iTMT MAE 0.96 mm (95%CI: 0.74–1.17) vs inter-expert MAE 1.20 mm (95%CI: 0.91–1.49) and in those with brain tumors (iTMT MAE 1.23 mm, (95%CI: 0.79–1.67) vs inter-expert MAE 0.92 mm, (95%CI: 0.52–1.31). For more details, refer to Supplementary Methods A4.

We performed acceptability testing by two validators on 2,950 TMs from randomly selected MRI scans stratified by age and sex to ensure accurate predictions (Fig. 2A). A 3rd tie-breaker validator reviewed disagreements to designate acceptability. Overall pipeline segmentation acceptability after tie-breaking was 98.3% with a high agreement (Gwet AC1 = 0.98)[36]. We qualitatively investigated the low number of unacceptable cases after tie-breaking (n = 111) and those with disagreement between reviewers (Fig. 2C). We identified that causes of such discrepancies were: low image resolution or corrupted MRI (35%), motion artifacts (<1%), and very small TM (<1%), with other cases having no clear cause (53%). For a detailed analysis of the MRI image quality and expected model sensitivity, see Supplementary Methods A5.

Given the existing literature supporting TMT as a biomarker for lean muscle mass and that iTMT derived from temporalis segmentation showed minimal inter-operator variability compared to cross-sectional area, we decided to focus on iTMT for further correlative analyses[37].

On univariate regression analysis within the ABCD cohort (age 8–13), Latino, Black, or Mixed race/ethnicity, if the family could not afford food in the past 12 months, if the family was born in the USA, lower household income, not having insurance, and lower levels of parent education were associated with increased iTMT. On multivariate regression analysis on the same cohort, statistically significant variables included Latino, Black, or Mixed race/ethnicity, if the family was born in the USA, and parent education (Supplementary Methods A11).

### Temporalis muscle normal reference growth charts

Given the high acceptance rate in the subsample, we applied iTMT to collected scans in the healthy cohort (n = 23,876) and plotted sex-specific iTMT by age from 4 to 30 (Fig. 3A). There was a median of 135 scans per age (IQR:81–299), with at least 60 scans for each year of age in the range of 4–30. For the 10,444 (43.8%) scans with documented race and ethnicity information, the distribution was comparable to that of the United States population nationally (Supplementary Table S7).

**Fig. 1 | Dataset summary and method overview. A** Aggregated dataset from 13 primary studies (total N = 23876 T1w MRI, see Supplementary Methods A7) with violin plot age distributions with annotated dataset size. Violin plots show the kernel density estimation of the distribution, with a boxplot overlay, with the median marked by a white dot, the interquartile range marked by the thick black bar, and the range by the thin black line. The darker color corresponds to the bigger dataset size. **B** iTMT performance compared to interobserver variability of Human Expert. The top panel: healthy, and the bottom: the Brain tumor cohort. Panel B1-Slice Selection MAE in mm (Human Expert for Healthy cohort ($n = 46$) MAE = 2.0 mm [IQR = 1.–2mm], iTMT MAE = 2.5 mm [IQR = 1–4.75 mm]; Human Expert for Brain tumor cohort ($n = 25$) Slice Selection MAE = 1.0 mm [IQR = 0–3 mm], iTMT MAE = 3.0 mm [IQR = 2–5 mm]). Panel B2-Dice (Human Expert for Healthy cohort Dice = 0.81 [IQR = 0.76–0.84], iTMT Dice = 0.84 [IQR = 0.81–0.87]; Human Expert for Brain tumor cohort Dice = 0.81 [IQR = 0.75–0.83], iTMT Dice = 0.81[IQR = 0.73–0.86]). Panel B3-iTMT MAE in mm (Human

Expert for Healthy cohort MAE = 1.20 mm [IQR = 0.66–1.96 mm], iTMT MAE = 0.96 mm [IQR = 0.4–1.53 mm]; Human Expert for Brain tumor cohort MAE = 0.92 mm [IQR = 0.5–1.4 mm], iTMT MAE = 1.23 mm [IQR = 0.55–1.9 mm]). TMT temporalis muscle thickness, MAE median absolute error, GAMLSS Generalized Additive Models for Location Scale and Shape. *P*-values were tested using the Mann–Whitney two-sided *U* test. Violin plots show the kernel density estimation of the distribution, with a boxplot overlay, with the median marked by a white dot, the interquartile range marked by the thick black bar, and the range by the thin black line. **C** Method overview. Step 1: Registration to age-specific MNI (Montreal Neurological Institute) template and manual check. Step 2: MRI T1w preprocessing: rescaling, z-normalization. Step 3: Slice selection via Dense Net model. Step 4: Segmentation via UNet prediction. Step 5: iTMT calculation using GAMLSS growth charts(error bands CI 2.5–97.5%). For more details on each step, please refer to the Methodology section. Source data are provided as a Source Data file.

We calculated sex-specific Generalized Additive Models for Location, Scale, and Shape (GAMLSS)[38] curves that explicitly estimate age-related variance, considering the known differences in development between males and females. (See Methods; Supplementary Materials A6)

The curves reveal distinct and, heretofore, undescribed phases of TM development among developing humans. For both sexes, there is steady TM growth through childhood, followed by a plateau, the timing of which is sex-specific (Fig. 3B).

On average, one iTMT centile point change translated to 0.296 mm. Given the MAE of 1.06 mm, we estimated that iTMT had precision within 6.0 centiles [95%CI: 5.3–7] for males and 7.1 centiles [95%CI: 6.1–8.5] for females (See Supplementary Methods A4).

Data used to produce the centiles charts is provided as a Source Data file. We provide an online centile calculator tool for community use at https://itmt-icsa.streamlit.app/. Sex-specific growth chart templates were generated for individual use (Supplementary Figs. S23–S24).

**iTMT and physiologic biomarkers**

We investigated how iTMT centile was associated with other clinical-physiologic health parameters in children aged 8–13 via the ABCD dataset[30]. Given the widespread use of height, weight, and BMI as a surrogate for body composition in children, we investigated their associations with iTMT. We found moderate correlations between

## A. "Acceptability" pipeline

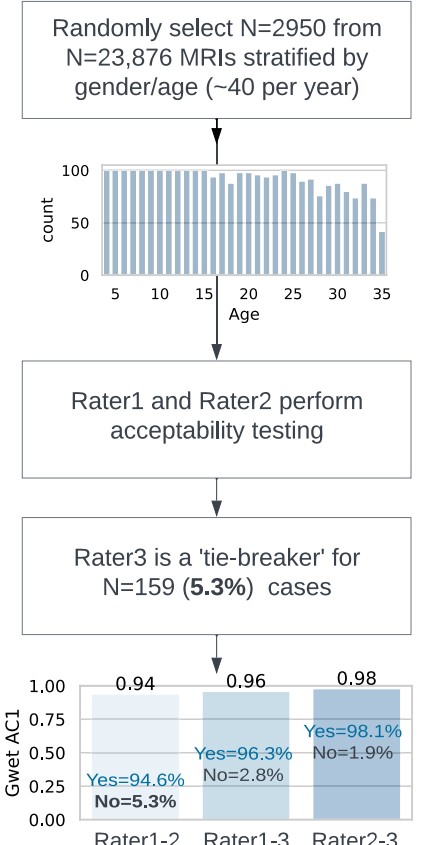

## B. Likert score summary

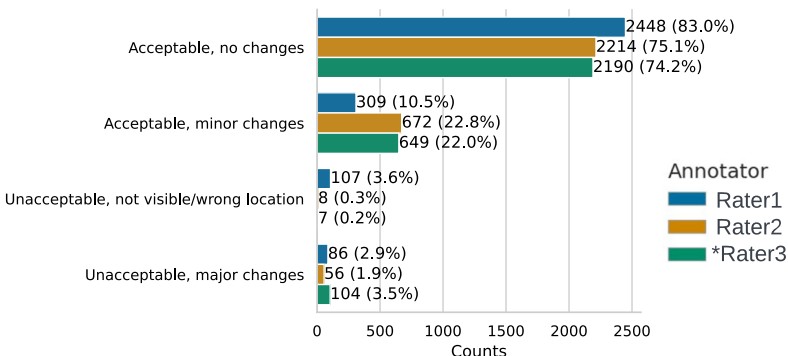

## C. Example of TM cases with disagreement

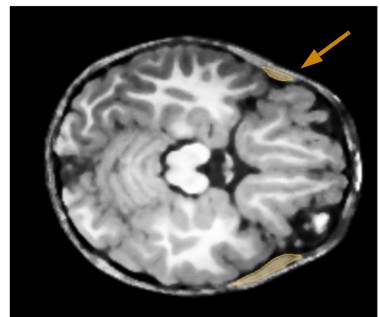 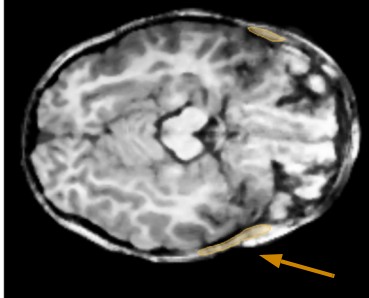

**Rater1:** Acceptable, minor changes
**Rater2:** Unacceptable, major changes
**Rater3:** Acceptable, minor changes

**Rater1:** Unacceptable, major changes
**Rater2:** Acceptable, minor changes
**Rater3:** Acceptable, no changes

**Fig. 2 | Acceptability testing pipeline. A** Overview of the data flow with Gwet AC1[36] for interobserver agreement. **B** Likert-type scores distribution among 3 annotators. Rater 1 and Rater 2 were two primary annotators, Rater 3 was the tiebreaker. **C** Two examples where primary annotators disagreed on the acceptability and a tiebreaker (Rater3) reviewed the cases. TM temporalis muscle. Source data are provided as a Source Data file.

iTMT and subjects' BMI and weight (Spearman's $\rho$: 0.63 and 0.63, respectively) and a low correlation between iTMT and subjects' height (Spearman's $\rho$: 0.379), suggesting that lean muscle mass measurement and sarcopenia cannot be directly estimated by BMI alone. In addition, we identified subgroups of patients with discordant iTMT and BMI, including a substantial number of patients with low iTMT and normal to high BMI ($N = 4304$, per CDC[39] defined reference ranges, see Supplementary Figs. S4–S5). These patients might be at high risk for the negative health effects of occult sarcopenia or sarcopenic obesity and benefit from therapeutic interventions, but more investigation is needed. Dehydroepiandrosterone (DHEA) has been positively associated with obesity[40,41], and its high levels in children can cause early puberty. Here, we found that higher iTMT was associated with increased levels of DHEA ($N = 13,256$, 52% Male, mean age 10 years, IQR = [9–11], Fig. 4A) along with higher caloric intake ($N = 2935$, 53% Male, mean age 11.4 years, IQR = [11–12]). Low testosterone levels were previously associated with increased fat mass and reduced lean mass in males[42]. Here, we found that iTMT was associated with increased testosterone levels ($N = 13256$, 52% Male, mean age 10 years, IQR = [9–11], Fig. 4A). Reduced levels of HDL-cholesterol is one of the risk factors for metabolic syndrome in adults[43], and we found out that it was positively associated with higher iTMT ($N = 626$, 55% Male, mean age 11.4, IQR = [11–12]).

### iTMT in pediatric patients with brain tumors

Sarcopenia and cancer cachexia[44] are components of physiologic frailty that have been linked to morbidity and early mortality and are prevalent in pediatric cancer survivors, particularly pediatric brain tumor survivors[12,45]. Pediatric gliomas, which include low-grade (LGG) and high-grade (DMG), are the most common solid primary CNS tumor in the pediatric age group[46]. In two cohorts of pediatric brain tumor patients, we found that the median iTMT centile was lower than in the healthy population (LGG: 33.3 centile [IQR = 6.6–52.9] and DMG: 36.19 centile [IQR = 10.3–62.1]) (Fig. 5; See Supplementary Methods A10 "Pediatric low-grade glioma (pLGG) and diffuse midline glioma (DMG)").

### Feasibility of longitudinal intra-patient iTMT tracking

The ability to reliably track iTMT longitudinally across an individual patient's growth and development, akin to classical height and weight growth charts, would be clinically useful. To assess the feasibility of intra-subject longitudinal measurements, we computed iTMT and the corresponding centile scores for a 23-year-old woman who underwent 30 days of consecutive MRI scans in two studies, each one year apart[47] (Fig. 6A). The standard deviation for iTMT was ±0.71 mm (Table S2, ±4.18 centiles) over all 30 consecutive days for year one and ±0.42 mm (Table S2, ±2.5 centiles) for days with motion artifact minimization; and ±0.38 mm (±2.23 centiles) for year two, indicating high precision and repeatability of intra-patient, longitudinal iTMT within 2-5 centiles, with a small dependence on scan quality.

In addition, we assessed longitudinal iTMT measurement feasibility in the ABCD cohort. We calculated iTMT for the 14,642 patient visits for children with two MRI scans taken within two years. We found that <2% of patients experienced dramatic iTMT centile change

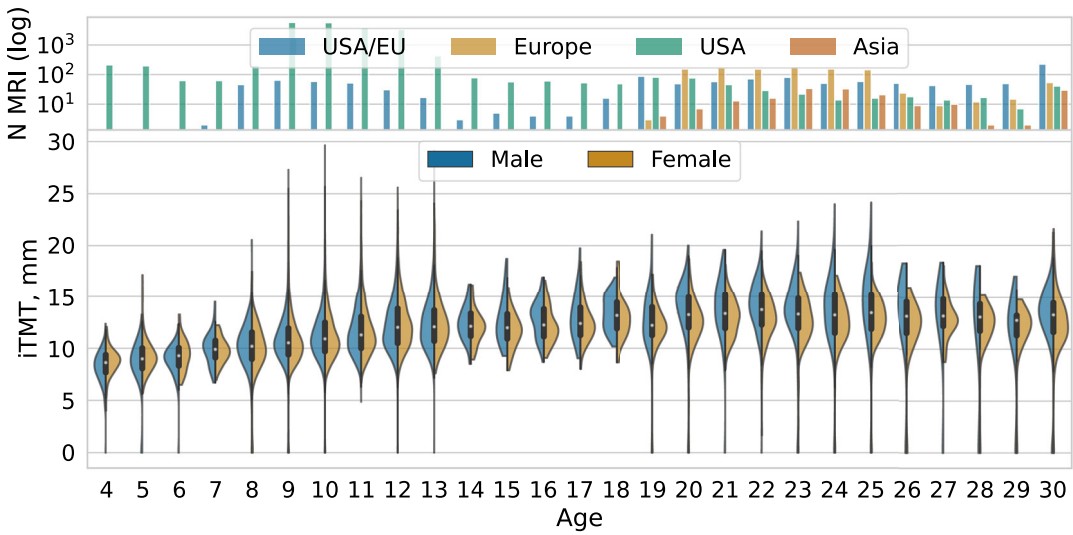

**A. Aggregated dataset for GAMLSS**

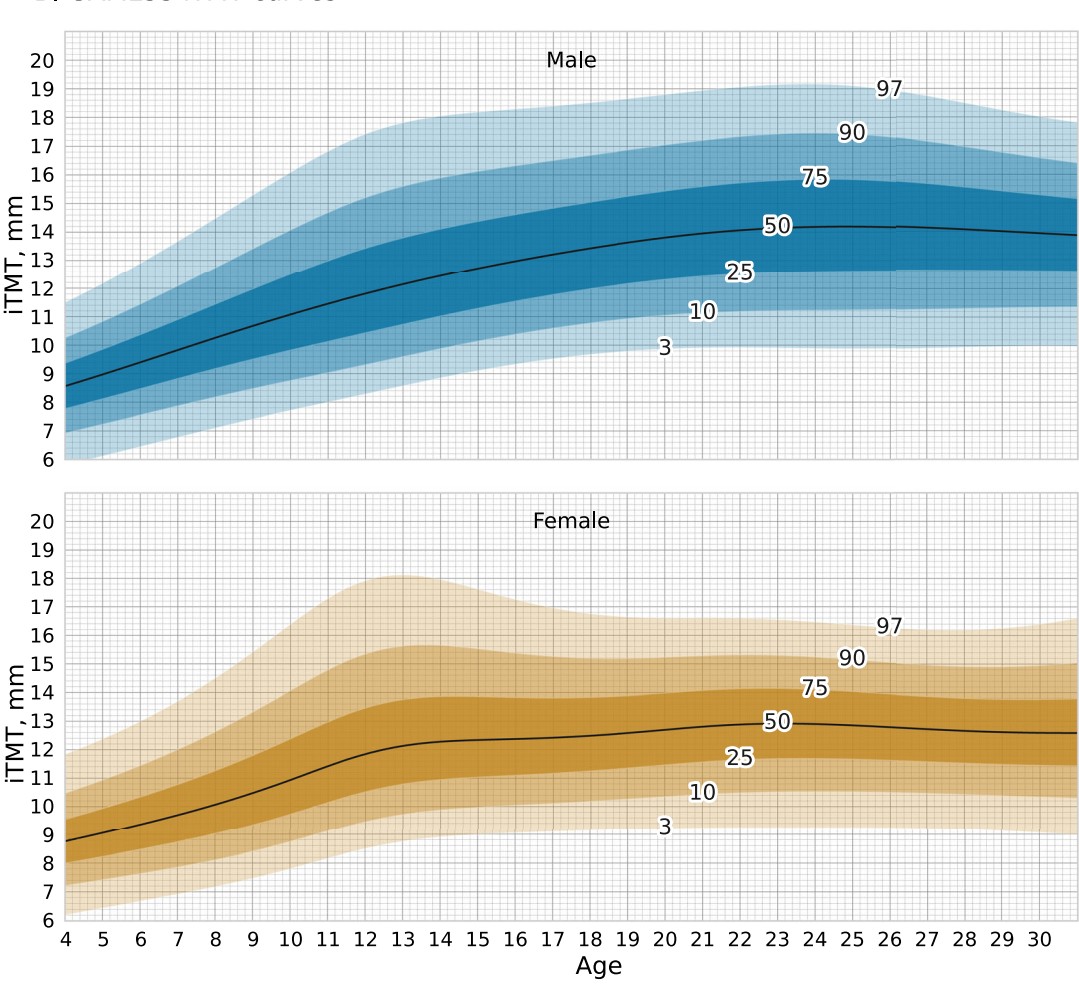

**B. GAMLSS iTMT curves**

(Δ > 50%)—which would be biologically implausible, and 15.8% experienced minor iTMT change (Δ > 25%), see Supplementary Fig. S4.

## Discussion

Leveraging the largest aggregated pediatric and young adult brain MRI dataset to-date, we developed and validated a multistage deep learning pipeline for accurate and reliable, automated temporalis muscle thickness calculation (iTMT) and generated practical, normal reference growth charts to track lean muscle mass in children through adulthood. Our study provides an unprecedented look into how the temporalis muscle changes during human development and defines sex-specific normal reference ranges that provide complementary information to BMI and anthropomorphics. Furthermore, the iTMT pipeline can be employed on routine T1-weighted MRI brain with or

**Fig. 3 | Temporalis Muscle Normal Reference Growth Charts. A** Sex-specific iTMT by age for the healthy cohort $N = 23,876$. Top panel: count plot demonstrates the geographical location of primary studies (scaled log $Y$-axis). Bottom panel: violin plot, raw predicted iTMT ranges split by biological sex. Violin plots show the kernel density estimation of the distribution, with a boxplot overlay, with the median marked by a white dot, the interquartile range marked by the thick black bar, and the range by the thin black line. The age group 30–35 was collapsed into one category 30+, due to the sparsity of the data available. The correlation between data size and the frequency of outliers (>3 standard deviations) is positive, indicating that larger datasets tend to have a greater number of outliers (The biggest

open source dataset ABCD[30] ($N = 18,949$ T1w MRIs) for the age group 8–13). See Supplementary Information A7 for details on demographics data. The mixed origins dataset was labeled as EU/USA. **B** iTMT normal reference growth charts with percentile lines for females (bottom) and males (top). We developed these charts by applying iTMT to 23,876 T1w MRI scans for patients aged 4–35 and creating growth centile curves estimated using GAMLSS. The age group 30–35 was collapsed into one category 30+ for curve stability. See Supplementary Figs. S23–S24 for CSA and iTMT Growth charts. GAMLSS Generalized Additive Models for Location Scale and Shape. Source data are provided as a Source Data file.

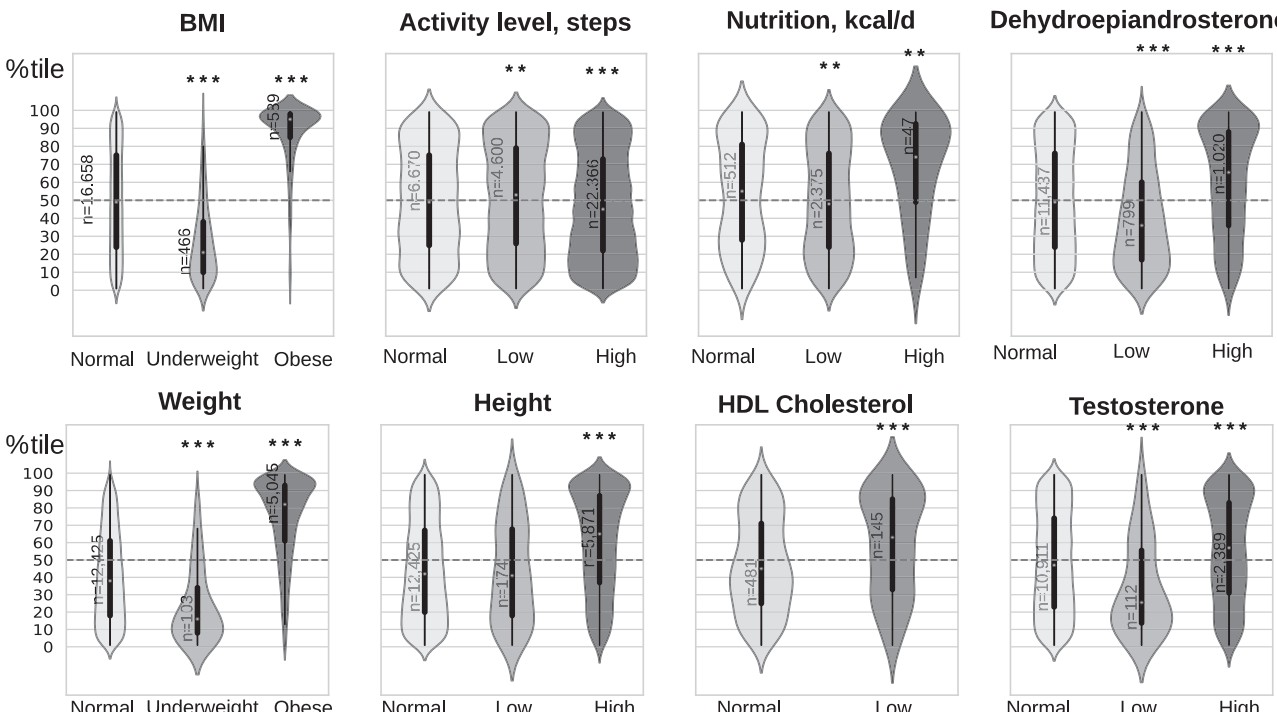

**Fig. 4 | The association between iTMT centile scores and patients' BMI[77], height/weight[77], activity levels[78], nutrition[79], dehydroepiandrosterone(DHEA)[80], cholesterol[81], and testosterone[82]** for patients aged 8–13 (Source: ABCD[30]). Violin plots show the kernel density estimation of the distribution, with a boxplot overlay, with the median marked by a white dot, the interquartile range marked by the thick black bar, and the range by the thin black line. Statistically significant groups (*$P < 0.05$, **$P < 0.01$, ***$P < 0.001$) that are labeled with an asterisk (*) were tested with two-sided Mann–Whitney $U$-test. For age- and sex-specific definitions of normal/high/low ranges for physical biomarkers, refer to Supplementary Methods A8. Source data are provided as a Source Data file.

without contrast and does not require specialized machinery and expertise nor radiation exposure. For the many people who receive routine MRI for symptoms or chronic illness, such as brain cancers and neurodevelopmental or degenerative disorders, iTMT may have immediate clinical utility without requiring extra resources, cost, or radiation exposure. Future applications of iTMT could be to track and predict morbidity through treatment and survivorship for people with serious illnesses. iTMT monitoring may reveal important physiologic states that merit intervention and have clinical utility in triaging patients for escalated care.

The iTMT pipeline utilizes a transparent and reproducible, stepwise approach to TMT calculation and demonstrates a high degree of accuracy, comparable to agreement between trained human experts. Furthermore, iTMT is accurate on a wide range of patients and T1-weighted scans with heterogeneous age, demographics, scanner parameters, image quality, contrast enhancement, and presence of brain pathology. It is notable that given the relatively small size of the temporalis muscle, small absolute variations in thickness measurement can translate into large changes in centile.

One centile point change translates on average to 0.08 mm, which requires precise iTMT quantification. Based on our investigations, iTMT likely has precision within ~6–7 centiles at baseline, with ~2–5 centiles for intra-patient longitudinal monitoring. We plan in the future to improve model performance by training on more challenging and variable cases, e.g., post-surgical MRI scans and those with ipsilaterally removed TM. In comparison to the single study of deep learning TM segmentation in adults[35], our pipeline achieved higher accuracy and better generalization across a range of brain pathologies and, critically, advances the approach to automated, scan-to-TMT calculation, which will lower the barrier to clinical adoption. We quantified TMT using the Feret diameter[48,49] (see Methods), a widely accepted method for measuring projections of a 3D object into 2D space in microscopy, obviating the need for manual measurement, which can be prone to interobserver error. Another advantage of using the Feret diameter is that it is not affected by muscle orientation and is robust, unlike a human subjective definition based on the marker location, which can be a concern with other methods of measurement.

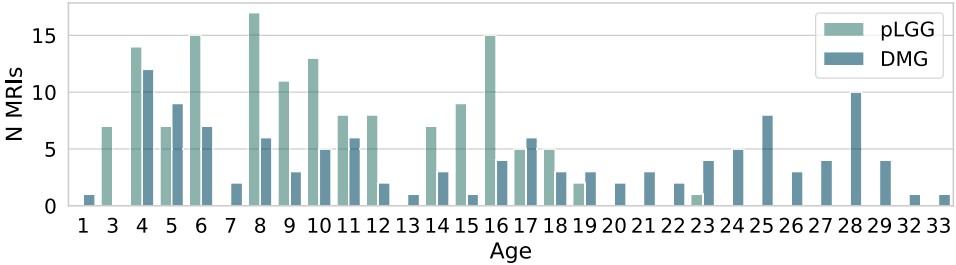

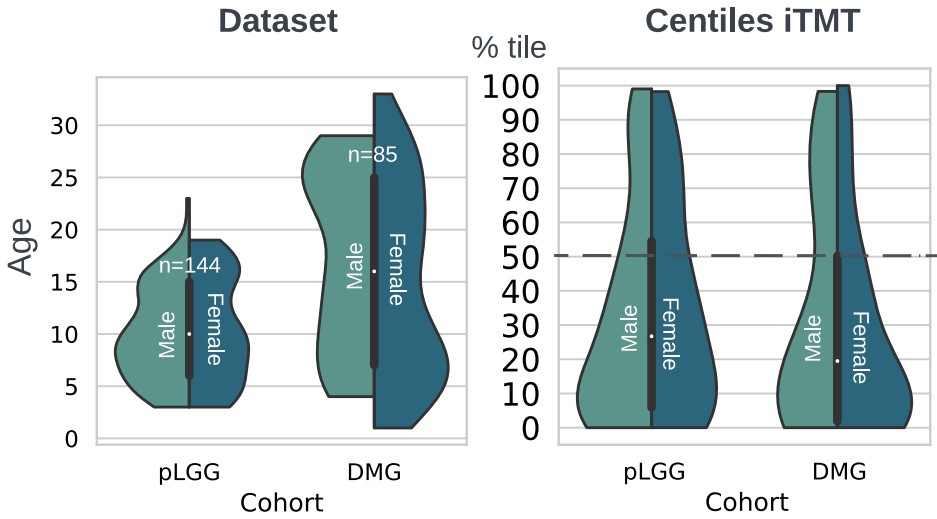

**Fig. 5 | iTMT in pediatric patients with brain tumors. A** Aggregated dataset from two studies with brain cancers (total $N = 229$ T1w MRI) age distribution. **B** iTMT and cancer cachexia for pLGG and DMG cohort. Source data are provided as a Source Data file.

By providing age- and sex-normalized metrics, centile scores enable trans-diagnostic comparisons between disorders that emerge at different stages of the lifespan. Future work will evaluate how combining CSA measurements with iTMT could further refine imaging-based sarcopenia assessment.

We demonstrated that iTMT was directionally related but not collinear to BMI. BMI is currently the most widely used method of identifying children and adolescents with excess adiposity and risk for the development of metabolic disease[23,50], but is a suboptimal screening tool for identifying individual patients at risk for physiologic frailty and mortality. In this work and others[51,52], TMT was associated with malnutrition, muscle wasting, and, in the case of pediatric cancer, survival, supporting its use as a health biomarker. We found that males had significantly higher TMT levels than females, which aligns with a study on lumbar abdominal muscle CSA and previous findings[35,53]. Around the onset of puberty, male TMT growth accelerates compared to female TMT and then plateaus in the early 20s (Fig. 3), perhaps due to a differential increase in testosterone and the effect on muscle mass[54].

For patients with brain tumors, a strong correlation was previously shown between the temporal muscle thickness (TMT)[55] and the skeletal muscle mass measured on cross-sectional abdominal CT, demonstrating the use of TMT as a surrogate marker for sarcopenia[37]. TMT has also been used for evaluating muscle mass and function among stroke patients[56] and for prognosis of patients with primary glioblastoma[33,35,57]. Future work will be required to tease out the incremental information gained by using iTMT in addition to BMI in terms of prognostication and identification of physiologic states.

This study has several important limitations. Firstly, our derived iTMT percentiles are largely composed of a United States-based cohort due to the lack of the open-source data and further study will be needed to determine the validity of iTMT percentile charts in international cohorts and under-represented minority populations. Race, ethnicity, and clinical data were lacking in most datasets, and our preliminary subgroup findings suggested that there may be small baseline differences in normal iTMT ranges across different datasets (Supplementary Fig. S29). For data with known demographic information, the sample distribution was roughly equivalent to the national census. The population-based iTMT thresholds used for associative analyses to represent the risk of sarcopenia and sarcopenic obesity are only hypothesis-generating and require validation with clinical outcomes. Which specific iTMT threshold dictates clinical sarcopenia may vary by the individual, based on their growth trajectory history, and requires further investigation.

Furthermore, TM segmentation of infant brain MRI is considerably more challenging than adult brain MRI due to the reduced tissue contrast, and TM is underdeveloped for children under 4. Therefore, iTMT is only applicable for children aged four and above.

Our study necessarily relies on cross-sectional imaging to develop reference curves, as there are no large longitudinal MRI datasets available for healthy children followed through young adulthood. While our study demonstrates the feasibility of applying iTMT to longitudinal patient data, further investigation is needed to validate iTMT as a longitudinal biomarker of growth and its associated clinical outcomes.

One potential failure mode is the presence of artifacts or poor image quality in the MRI images, which can affect the accuracy of the

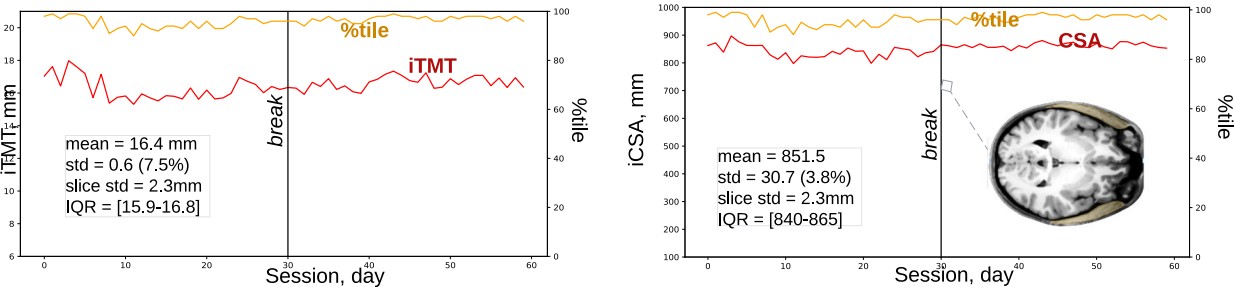

## A. iTMT and iCSA intrapatient stability

## B. Longitudinal intra-patient iTMT assesment

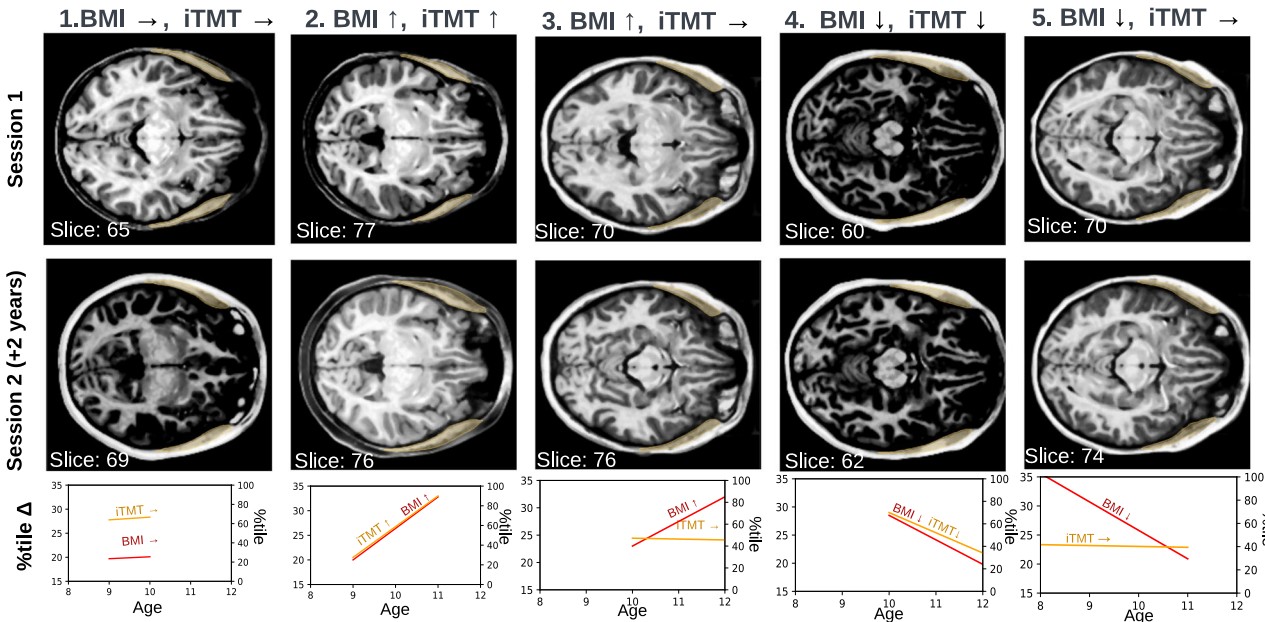

**Fig. 6 | Longitudinal intra-patient iTMT assessment. A** The participant (Female, 23 y.o.) underwent daily testing for two studies of 30 consecutive days with one year in between[47]. Left: iTMT and centile measurement computed for daily T1w scans; right: CSA and centile scores with an overlay of predicted iTMT mask T1w MRI taken on the 30th day. The mean iTMT for 1st session (duration of 30 days) was 16.3 mm ± 0.71 mm; the mean iTMT for both studies was 16.4 mm ± 0.6 mm. The mean slice selection error is 2.7 mm. **B** Feasibility of tracking intra-patient changes (dataset: ABCD[30]). On the top right of each MRI, the age and biological sex is displayed, and on the bottom left information about the predicted slice by the first stage. (1) BMI stable, TMT stable; (2) BMI increased, TMT increased; (3) BMI increased, TMT stable, (4) BMI increased, TMT decreased; (5) BMI decreased, TMT stable. BMI body mass index, TMT temporalis muscle thickness. Source data are provided as a Source Data file.

segmentation. In addition, a manual or automated MRI quality check is recommended prior to use of iTMT, which, in this study, was only evaluated on MRIs for which the temporalis muscle was sufficiently visible for a human to segment. The performance of our model may be affected by the size and location of the tumor or surgical cavity within the brain. There were notably two outliers in the brain tumor cohort with low Dice scores (Fig. 1B). We investigated these cases and found that the reason was an abnormal brain anatomy with high ventricular volume (see Supplementary Fig. S10) and concluded that iTMT would benefit from additional re-training on edge cases to improve reliability further. Another potential pipeline limitation is the failure of the precise registration alignment to the age-specific template. Specifically, brains with high anatomical deformities are at high risk due to structural differences in comparison to a normal brain. Lastly, during the curation of the open-source datasets, multiple studies were excluded due to specific MRI anonymization techniques that corrupted the region of the temporalis muscle. This highlights the need to review scans post-anonymization process prior to implementing the iTMT pipeline.

In conclusion, we leveraged a large, aggregated dataset of pediatric brain MRI and a multistage deep learning pipeline to develop a fully automated TMT calculation and generate TMT growth charts for children through young adulthood. The iTMT pipeline, coupled with these growth charts, enables individualized tracking of patient lean muscle mass status to inform clinical decision-making and interventions.

## Methods

The overall pipeline is shown in Fig. 1C. In summary, we pre-processed MRI scans by applying registration and image normalization methods. Next, we trained a slice selection model for automated top orbital roof slice selection and a separate model for auto-segmentation of the TM. To measure TMT and CSA, we use Python minimum Feret diameter implementation[48]. Measurements were conducted at each side at the level of the superior orbital roof (cranio–caudal landmark) perpendicular to the long axis of the temporal muscle. We followed the previous study by Steindl et al.[58], measuring TMT values on both sides and dividing them by two to calculate the mean TMT values for each patient and reduce dental- or oral-related muscle changes.

## Datasets

We curated 23,852 MRIs from 13 datasets (ABCD[30], ABIDE[59], AOMIC[60], Baby Connectome[61], Calgary[62], ICBM[63], IXI[64], NIMH[31], PING[65], Pixar[32], SALD[66], NYU2(CoRR)[67], Healthy Adults;[68] 90% United States; 52% male; 26% White, 6% Hispanic, 5% Black, 2% Asian, 4% Mixed, 1% Other, 56% Unknown; see Supplementary Methods A7).

## Image acquisition and registration

Scans that were downloaded in native DICOM format were converted to NifTI via Python Pydicom package[69]. Next, scans were co-registered to MRI age-dependent T1-weighted asymmetric brain atlases, generated from the NIH-funded MRI Study of Normal Brain Development (hereafter, NIHPD, for NIH pediatric database[70]) with rigid registration using SlicerElastix[71] (Elastix generic rigid preset). For the training and testing cohort, a manual quality check was conducted on each scan to assess that the anterior and posterior commissures were co-planar and that there was no significant lateral tilt prior to annotation[70].

## Temporalis muscle annotation

We recruited five annotators for ground-truth TM segmentation and one board-certified radiologist with eight years of expertise to perform validation and/or correction of the annotations. Annotators received dedicated training on the protocol for TM segmentation (Supplementary Methods A1).

## Image preprocessing

Following co-registration and annotation (if performed), MRI images were rescaled to 1-mm isotropic voxel size using itk-elastix Python package[72]. We then normalized MRI images using the Z-Score method[73], performed median filtering, removed background pixels using Otsu filtering, and standardized the intensity scale. For more details, please refer to the study code repository https://doi.org/10.5281/zenodo.8428986.

## Slice selection model

We trained the DenseNet[74] regression model for the automated top orbital roof slice selection from brain MRI. We generated maximum intensity projection slices with 5 mm thickness $256 \times 256$ and the corresponding label that encoded the offset from the target slice. We trained the model with $N = 23,680/5920$ images before pseudo-labeling and $45,695/5920$ after pseudo-labeling training/validation images. We added data augmentations, including 10-degree rotations and width/height shifts. We trained the model using Adam optimizer for 30 epochs with batch size 64 and mean squared error (MSE) loss with an initial learning rate 1e−4. We set up the learning rate scheduler to reduce on a plateau with starting learning rate = 5e−4 and used a 1x Nvidia A6000 for training with TensorFlow v.2.10, Python v.3.9.

## Segmentation model

For the segmentation, we trained a 2D UNet for 30 epochs with batch size 4 with an initial learning rate 5e−4 and the same strategy for the learning rate decreased as described above. We upscaled images into $512 \times 512$ and used five downsampling/upsampling modules. We added data augmentations, including 10 deg rotations, width and height shifts, horizontal flips, and zooming. We use Focal Tversky Loss proposed by Abraham et al.[75]. Since UNet tends to have bad predictions around edge areas, we create a major voting post-processing step. Each TM muscle was predicted four times, and tiles were overlapped so that each pixel was voted at least three times (see Supplementary Information A4 for a comparison of the model postprocessing methods).

## Temporalis muscle thickness

For automated TMT measurement, we used the implementation of Feret diameter in Python[48]. Feret diameter is the distance between the two parallel planes restricting the object perpendicular to that direction[49] (see Supplementary Methods A3).

## Generalized additive models for location scale and shape (GAMLSS)

The iTMT curve fitting and chart generation was performed using the GAMLSS function in R (version 4.2.2, RStudio 2022.12.0), which has been used in CDC and WHO growth charts and uses the data generated from iTMT along with demographical variables such as age and sex (Fig. 3B, Supplementary Methods A6). To test whether our model's reliability was skewed, we performed a leave-one-study-out (LOSO) analysis (Supplementary Methods A9). In the context of the present study, we used the Bayesian information criterion (BIC) to assess the goodness-of-fit of GAMLSS models making different assumptions about the form of the phenotypic distributions[76].

## Physiologic biomarkers

For the association of iTMT and patient characteristics, we used the ABCD dataset (ages 8–13)[30] under Data Use Agreement. We analyzed the association between temporalis thickness and patients' BMI, height/weight levels, Dehydroepiandrosterone (DHEA), daily caloric intake, activity, and testosterone using with two-sided Mann–Whitney $U$-test (see Supplementary Methods A8 for the normal/low/healthy range definitions and Supplementary Methods A2 for the head-circumference iTMT adjustment). Statistical significance was set at a two-tailed $p$-value of <0.05. All analyses were performed using Python SciPy package.

## Reporting summary

Further information on research design is available in the Nature Portfolio Reporting Summary linked to this article.

## Data availability

All data supporting the findings described in this manuscript are available in the article, in the Supplementary Information, and from the corresponding author upon request. Specifically, all public datasets used for this study can be found via the description and links in Supplementary Information A7. The BCH brain tumor dataset contains private hospital data that is controlled due to privacy concerns. Access to the derived dataset will be considered upon request to the corresponding author (Benjamin H. Kann, M.D., email: Benjamin_Kann@dfci.harvard.edu, timeframe for response 2 weeks). All model parameters, weights, and details are provided in Supplementary Information A7. Source data are provided with this paper.

## Code availability

The model training and testing code is available at https://doi.org/10.5281/zenodo.8428986. iTMT and iCSA centile calculator is available at https://itmt-icsa.streamlit.app/.

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

## Acknowledgements

The authors acknowledge Ariana Familiar and research staff at the Children's Brain Tumor Network (CBTN) for facilitating data used for this study. We acknowledge the invaluable contribution to this work made by several openly shared MRI datasets: the Adolescent Brain Cognitive Development study (ABCD), Autism Brain Imaging Data Exchange (ABIDE), Amsterdam Open MRI Collection (AOMIC), Baby Connectome, Calgary Preschool MRI study, International Consortium for Brain Mapping (ICBM), IXI study, NIMH Pediatric MRI dataset, PING Data Resource, Pixar (OpenfMRI database), Southwest University Adult Lifespan Dataset (SALD), NYU2 Consortium for Reliability and Reproducibility, NIMH Healthy Research Volunteer Study, and 28andMe. The authors acknowledge financial support from NIH (HA: NIH-USA U24CA194354, NIH-USA U01CA190234, NIH-USA U01CA209414, and NIH-USA R35CA22052; B.H.K.: Botha-Chan Low Grade Glioma Consortium, NIH-USA U54 CA274516 and NIH-USA K08DE030216-01), and the European Union—European Research Council (HA: 866504). K.L. is funded by the National Institutes of Health Loan Repayment Program L40 CA264321. All analyses and conclusions in this manuscript are the sole responsibility of the authors and do not necessarily reflect the opinions or views of the clinical trial investigators, the NCTN, or the NCI.

## Author contributions

Conceptualization and Study Design: A.Z., B.H.K.; Data collection/curation: A.Z., B.H.K., T.M.C., T.Y.P., D.A.H.K., K.X.L., Z.Y., S.V., R.B.C., Y.R., A.J., A.M.F., J.H., H.H.; Investigation: A.Z., B.H.K., A.S., Z.Y.; Code, Software: A.Z., Z.Y., Y.R.; Methodology, Formal Analysis, Visualizations (Figures): A.Z., B.K.; Data Interpretation: A.Z., B.H.K.; Manuscript Writing—original draft: A.Z., B.H.K.; Manuscript Writing—review & editing: A.Z., K.X.L., A.S., Z.Y., P.J.C., V.B., Y.R., A.J., J.H., H.H., J.L., S.V., R.B.C., A.N., R.H.M., A.C.R., S.M., T.M.C., D.A.H.K., T.Y.P., H.J.W.L.A., B.H.K.; Project administration: B.H.K., H.J.W.L.A.; Resources: B.H.K., H.J.W.L.A., T.Y.P.; Supervision: B.H.K., H.J.W.L.A. All authors have substantively revised the

work, reviewed the manuscript, have approved the submitted version, and have agreed to be personally accountable for their contributions.

## Ethical approval and informed consent

For the BCH dataset, all data was de-identified for the purposes of the study and deemed non-human subject research per the Federal Regulation 45 CFR 46 "Protection of Human Subjects". As such, waiver of consent was granted by the Institutional Review Board of Dana-Farber/Harvard Cancer Center, Protocol #13-055, as secondary use research. The rest of the datasets (ABCD[30], ABIDE[59], AOMIC[60], Baby Connectome[61], Calgary[62], ICBM[63], IXI[64], NIMH[31], PING[65], Pixar[32], SALD[66], NYU2(CoRR)[67], Healthy Adults[68], 28andme[47]) were anonymized and not collected by the investigators, in which case the work is classified as non-human research.

## Competing interests

The authors declare no competing interests.
