## [Peer Review File · Nature Communications]

Automated Temporalis Muscle Quantification and Growth Charts for Children Through AdulthoodEditorial Note: This manuscript has been previously reviewed at another journal that is not operating a transparent peer review scheme. This document only contains reviewer comments and rebuttal letters for versions considered at *Nature Communications*.

REVIEWER COMMENTS

Reviewer #1 (Remarks to the Author):

Well done to the authors for a comprehensive response to my previous comments - thank you! I only have two additional follow-up comments for the authors to consider as below

1. Comment: It is important for the authors to evaluate the comparability of data from the 13 sources and quantify the variability in TMT across the 13 data sources. This is key for establishing whether the derived growth charts were influenced by specific data or not. They should make a case for combining the data from the 13 sources to develop a unified growth chart for TMT. The authors should quantify/state and justify how much variability in TMT across the 13 study data is acceptable for pooling together the data for inclusion in the pooled sample for constructing TMT growth charts.

Response: Thank you for your comment highlighting the importance of evaluating the comparability and variability of data from the 13 sources in TMT. To address this comment, we have performed several analyses, including a leave-one-study-out (LOSO) analysis (Supplementary material A.13 Leave-one-study-out analysis). This approach allows us to quantify the variability and assess the robustness of the derived growth charts, further strengthening the validity and reliability of our findings. We observed minor differences between median LOSO GAMLSS fitted curves, indicating that the overall curves weren't overly influenced by a particular dataset (Figure S33).

In regards, to justification for pooling our data, in our study, pooling data from multiple sources was a necessity to capture a complete representation of TMT growth patterns from young children through adulthood, as datasets only included subjects within a smaller subset age ranges. However, we also recognize that there can be variability in TMT measurements across different studies due to various factors like patient selection criteria. We have now conducted an analysis of dataset variability, graphically plotting the iTMT medians and distributions of all 13 datasets by age (Figure S28). The plots demonstrate overall low variability across datasets at each age range. It should also be noted that in the pooled analyses, those datasets with larger sample sizes will automatically be given more weight, and so smaller datasets with presumably more variability and heterogeneity will be down-weighted automatically.

NEW COMMENT: Thanks to the authors for these additional analyses which is very informative. In addition to the leave one out study approach that the authors have done, it would be helpful to see the differences in the leave one out study approach if these were expressed as a standardised metric that a

kin to a z-score. For example, whilst the differences might appear small when compared to fitted centiles on the raw scale, this might be different when expressed in relative terms and expressing these differences as a z-score ensures that the differences also take into account age. A 1cm difference in early ages is not the same as 1 cm difference in later ages due to the increasing variability of TMT according to age. One way to do this would be to consider intervals of age say <2 years, 2-4 years etc and for each age interval, calculate the mean TMT and SD for that age range overall using all data, then for each of the 13 data sources, also calculate the mean TMT for each specific age interval. A standardised metric that is similar to a z-score can then be computed ie

Observed = mean TMT for a specific data source for a specific age range eg <2 years

Expected = mean TMT using data from all 13 data sources for a specific age range eg <2 years

SD = SD for TMT using data from all 13 data sources for a specific age range eg <2 years

Repeat this for other age ranges such as 2-4yrs, 4-6yrs and then can plot to see how large these differences are and whether they differ/vary by age.

2. Comment: Fig 1B – interobserver variability is best quantified using approaches such as the Bland-Altman plot which shows agreement between iTMT and human expert and clearly shows the variability. The goal for this is to quantify agreement and therefore quantifying the level of precision is more meaningful than using the MAE and IQR. The authors should show / comment on the comparison of iTMT and human expert across different ages. Were there differences observed as a function of age in terms of agreement?

Response: Thank you for this important feedback. We agree that Bland-Altman plots would be an ideal way to demonstrate agreement and provide a clearer understanding of the variability. In response to the feedback, we have now included a Bland-Altman plot in Figure S15, which shows the agreement between iTMT and the human expert. This plot allows for a visual representation of the agreement and provides a more precise assessment of the level of agreement between the two methods.

Additionally, we have included the histogram of agreement between the model and human expert across different ages, considering the potential differences observed as a function of age in terms of agreement (Figure S14). We found that both inter-expert and model- expert agreement improved with age, likely owing to larger and more robustly identifiable temporalis muscles in adolescents and young adults.

NEW COMMENT: Thank you for the additional analysis and Bland Altman. Check numbering of figures in the Supplementary file as they seem off. For example Bland-Altman plot is S14 not S15 as referred to here. I did not see the histogram figure S14

Reviewer #2 (Remarks to the Author):

The authors have incorporated most of my recommendations and comments in the new revision. I only have minor comments on the revised manuscript.

For the caption for Figure 3B, the panels are listed as being left and right, when they are actually upper and lower panels.

In Section 2.3 on page 12, there is a callout for Supplemental Material A7 and Table S7. However it appears that the relevant table is S5. It appears that the callouts no longer match the corresponding figures and tables in other areas of the Supplementary Material as well.

There is a new section in the Supplemental Materials about iTMT and social determinants of health. The authors report that ethnicity, family birth in the US, household income, and parent education were statistically associated with iTMT. Was this based on the univariable or multivariable analysis? If it was based on the univariable analysis, insurance status and food affordability also seem to be significantly associated with iTMT. However, if it was based on the multivariable analysis, the household income does not meet a significance threshold of $p < 0.05$. I would also recommend commenting on why higher income and parent education would be associated with lower iTMT (which is purportedly an indicator of poorer health). The text of this section also calls out Table S16 when the relevant table appears to be Table S14.

REVIEWER COMMENTS

Reviewer #1 (Remarks to the Author):

1. Well done to the authors for a comprehensive response to my previous comments - thank you! I only have two additional follow-up comments for the authors to consider as below

1. **Comment:** It is important for the authors to evaluate the comparability of data from the 13 sources and quantify the variability in TMT across the 13 data sources. This is key for establishing whether the derived growth charts were influenced by specific data or not. They should make a case for combining the data from the 13 sources to develop a unified growth chart for TMT. The authors should quantify/state and justify how much variability in TMT across the 13-study data is acceptable for pooling together the data for inclusion in the pooled sample for constructing TMT growth charts.

Response: Thank you for your comment highlighting the importance of evaluating the comparability and variability of data from the 13 sources in TMT. To address this comment, we have performed several analyses, including a leave-one-study-out (LOSO) analysis (Supplementary material A.13 Leave-one-study-out analysis). This approach allows us to quantify the variability and assess the robustness of the derived growth charts, further strengthening the validity and reliability of our findings. We observed minor differences between median LOSO GAMLSS fitted curves, indicating that the overall curves weren't overly influenced by a particular dataset (Figure S33).

In regards, to justification for pooling our data, in our study, pooling data from multiple sources was a necessity to capture a complete representation of TMT growth patterns from young children through adulthood, as datasets only included subjects within a smaller subset age ranges. However, we also recognize that there can be variability in TMT measurements across different studies due to various factors like patient selection criteria. We have now conducted an analysis of dataset variability, graphically plotting the iTMT medians and distributions of all 13 datasets by age (Figure S28). The plots demonstrate overall low variability across datasets at each age range. It should also be noted that in the pooled analyses, those datasets with larger sample sizes will automatically be given more weight, and so smaller datasets with presumably more variability and heterogeneity will be down-weighted automatically.

NEW COMMENT: Thanks to the authors for these additional analyses which is very informative. In addition to the leave-one-out study approach that the authors have done, it would be helpful to see the differences in the leave-one-out study approach if these were expressed as a standardized metric that kin to a z-score. For example, whilst the differences might appear small when compared to fitted centiles on the raw scale, this might be different when expressed in relative terms and expressing these differences as a z-score

ensures that the differences also take into account age. A 1cm difference in early ages is not the same as 1 cm difference in later ages due to the increasing variability of TMT according to age. One way to do this would be to consider intervals of age say <2 years, 2-4 years etc and for each age interval, calculate the mean ToMT and SD for that age range overall using all data, then for each of the 13 data sources, also calculate the mean TMT for each specific age interval. A standardised metric that is similar to a z-score can then be computed ie

Observed = mean TMT for a specific data source for a specific age range eg <2 years

Expected = mean TMT using data from all 13 data sources for a specific age range eg <2 years

SD = SD for TMT using data from all 13 data sources for a specific age range eg <2 years

Repeat this for other age ranges such as 2-4yrs, 4-6yrs and then can plot to see how large these differences are and whether they differ/vary by age.

Author Response:

Thank you for taking the time to review our paper and provide such insightful feedback thoroughly. Getting your perspective and incorporating your suggestions to strengthen our work was extremely valuable. We found your feedback constructive, discerning, and helpful in pushing our thinking forward.

In response to your request, we have added Error! Reference source not found. to demonstrate differences in mean and standard deviation in temporalis muscle thickness (in mm) by age and data source. Plots reveal variations across data sources in standardized mean TMT within given developmental periods. The raw data table is available as a Supplementary table for download. We have also added the following verbatim to “Supplementary material”, page 25, paragraph 2 :

“To measure dataset variability, we conduct Leave-one-out analysis (A.13) and mean analysis (Error! Reference source not found. and Error! Reference source not found.). The greater variation in standardized mean TMT scores observed in some data sources and age groups can be attributed to smaller subset sample sizes. With fewer subjects in a given age range for certain sources, the mean TMT calculation may have relied on just one or two individuals' scores.”

2. Comment: Fig 1B – interobserver variability is best quantified using approaches such as the Bland-Altman plot which shows agreement between iTMT and human expert and clearly shows the variability. The goal for this is to quantify agreement and therefore quantifying the level of precision is more meaningful than using the MAE and IQR. The authors should show / comment on the comparison of iTMT and human expert across different ages. Were there differences observed as a function of age in terms of agreement?
Response: Thank you for this important feedback. We agree that Bland-Altman plots would be an ideal way to demonstrate agreement and provide a clearer understanding of the variability. In response to the feedback, we have now included a Bland-Altman plot in

Figure S15, which shows the agreement between iTMT and the human expert. This plot allows for a visual representation of the agreement and provides a more precise assessment of the level of agreement between the two methods.

Additionally, we have included the histogram of agreement between the model and human expert across different ages, considering the potential differences observed as a function of age in terms of agreement (Figure S14). We found that both inter-expert and model- expert agreement improved with age, likely owing to larger and more robustly identifiable temporalis muscles in adolescents and young adults.

NEW COMMENT: Thank you for the additional analysis and Bland Altman. Check numbering of figures in the Supplementary file as they seem off. For example Bland-Altman plot is S14 not S15 as referred to here. I did not see the histogram figure S14

Response:

Thank you for catching the incorrectly numbered figure references in the response letter. You are correct that the Bland-Altman plot is labeled Figure S14, not S15. And we apologize for mislabeling Figure S14 as a histogram in the previous response - it is a scatter plot with a regression line. We have carefully reviewed the numbering and labels for all supplementary figures and tables and corrected any issues to ensure they match the referenced callouts in the main text and legends.

Reviewer #2 (Remarks to the Author):

1. The authors have incorporated most of my recommendations and comments in the new revision. I only have minor comments on the revised manuscript.

For the caption for Figure 3B, the panels are listed as being left and right, when they are actually upper and lower panels.

Author Response:

Thank you for reviewing our manuscript and providing extremely insightful and constructive feedback. We found your rigorous review of our work to be very helpful and believe it has significantly strengthened the final manuscript. Thank you for catching this mistake in Figure 3B- we have updated the caption to indicate "upper panel" and "lower panel" correctly.

2. In Section 2.3 on page 12, there is a callout for Supplemental Material A7 and Table S7. However it appears that the relevant table is S5. It appears that the callouts no longer match the corresponding figures and tables in other areas of the Supplementary Material as well.

There is a new section in the Supplemental Materials about iTMT and social determinants of health. The authors report that ethnicity, family birth in the US, household income, and parent education were statistically associated with iTMT. Was this based on the univariable or multivariable analysis? If it was based on the univariable analysis, insurance status and food affordability also seem to be significantly associated with iTMT. However, if it was based on the multivariable analysis, the household income does not meet a significance threshold of $p < 0.05$. I would also recommend commenting on why higher income and parent education would be associated with lower iTMT (which is purportedly an indicator of poorer health). The text of this section also calls out Table S16 when the relevant table appears to be Table S14.

Author Response:

We have reviewed the entire paper and supplementary files again to ensure that all figures, tables, and section references are now accurate. Regarding the new section on social determinants of health, we appreciate the reviewer raising an excellent point - we should have specified whether the significant associations were based on univariable or multivariable analyses. Thank you for catching that only household income met the multivariable model's $p < 0.05$ significance threshold. We have updated the text to indicate this more clearly, page 8, paragraph 4:

“On univariate regression analysis within the ABCD cohort (age 8 – 13), race/ethnicity, if the family was born in the USA, household income, insurance status, and parent education were associated with increased iTMT. On multivariate regression analysis on the same cohort statistically significant were variables race/ethnicity, if the family was born in the USA, household income and parent education (Supplement A15, Table S16)”

It is interesting that higher parent education was associated with lower iTMT. We believe the elucidating the reasons behind this will require further study and information outside the scope of the datasets used in this study. One possibility, is that given the known (though imperfect) correlation between BMI and iTMT, that increasing iTMT in lower educated households may be reflective of increasing weight/BMI in general. It is also likely that there are additional social factors at play that are not capture in the ABDC dataset. We have added commentary on this with the following sentences to Supplementary material, page 47, paragraph 1:

“This observation could be related to the disparities found across different groups and socioeconomic factors. This could be, in part, reflective of increased BMI observed in these cohorts or other social factors influencing nutrition and/or exercise habits^{1,2}, though these hypotheses require further investigation.”

References:

1. Seum, T., Meyrose, A.-K., Rabel, M., Schienkiewitz, A. & Ravens-Sieberer, U. Pathways of Parental Education on Children's and Adolescent's Body Mass Index: The Mediating Roles of Behavioral and Psychological Factors. *Frontiers in Public Health* **10**, (2022).
2. Ogden, C. L. Prevalence of Obesity Among Youths by Household Income and Education Level of Head of Household — United States 2011–2014. *MMWR Morb Mortal Wkly Rep* **67**, (2018).

REVIEWERS' COMMENTS

Reviewer #1 (Remarks to the Author):

Thanks to the reviewers for responding to the remaining questions from my second review. I have no further comments

Reviewer #2 (Remarks to the Author):

The authors report having reviewed the numbering of the tables, figures, and their respective callouts. However, the 2 specific errors I described in my comments remain uncorrected.

The changes to the section about the social determinants of health seem to have made it more incorrect. To avoid further confusion, I have edited the new paragraph based on how I am interpreting Table S14.

“On univariate regression analysis within the ABCD cohort (age 8 – 13), Latino, Black, or Mixed race/ethnicity, if the family could not afford food in the past 12 months, if the family was born in the USA, lower household income, not having insurance, and lower levels of parent education were associated with increased iTMT. On multivariate regression analysis on the same cohort, statistically significant variables included Latino, Black, or Mixed race/ethnicity, if the family was born in the USA, and parent education (Supplement A15, Table S14).”

Reviewer #1 (Remarks to the Author):

Thanks to the reviewers for responding to the remaining questions from my second review. I have no further comments.

Author Response:

Thank you for reviewing our manuscript and providing extremely insightful and constructive feedback. We found your rigorous review of our work to be very helpful and believe it has significantly strengthened the final manuscript.

Reviewer #2 (Remarks to the Author):

The authors report having reviewed the numbering of the tables, figures, and their respective callouts. However, the 2 specific errors I described in my comments remain uncorrected.

The changes to the section about the social determinants of health seem to have made it more incorrect. To avoid further confusion, I have edited the new paragraph based on how I am interpreting Table S14.

“On univariate regression analysis within the ABCD cohort (age 8 – 13), Latino, Black, or Mixed race/ethnicity, if the family could not afford food in the past 12 months, if the family was born in the USA, lower household income, not having insurance, and lower levels of parent education were associated with increased iTMT. On multivariate regression analysis on the same cohort, statistically significant variables included Latino, Black, or Mixed race/ethnicity, if the family was born in the USA, and parent education (Supplement A15, Table S14).”

Author Response:

Thank you for taking the time to review our paper and providing feedback thoroughly. Getting your perspective and incorporating your suggestions to strengthen our work was extremely valuable. In response to your suggestion, we have added the text you drafted to “Results” section, last paragraph, page 9, which we think reads clearly now, and accurately describes the data:

“On univariate regression analysis within the ABCD cohort (age 8 – 13), Latino, Black, or Mixed race/ethnicity, if the family could not afford food in the past 12 months, if the family was born in the USA, lower household income, not having insurance, and lower levels of parent education were associated with increased iTMT. On multivariate regression analysis on the same cohort, statistically significant variables included Latino, Black, or Mixed race/ethnicity, if the family was born in the USA, and parent education (Supplement A15, Table S16).”